# Bivariate genome-wide association meta-analysis of pediatric musculoskeletal traits reveals pleiotropic effects at the *SREBF1/TOM1L2* locus

Carolina Medina-Gomez [1,2,3], John P. Kemp [4,5], Niki L. Dimou[6,7], Eskil Kreiner[8], Alessandra Chesi[9], Babette S. Zemel[10,11], Klaus Bønnelykke[8], Cindy G. Boer [1], Tarunveer S. Ahluwalia [8,12], Hans Bisgaard[8], Evangelos Evangelou[7,13], Denise H.M. Heppe[2,3], Lynda F. Bonewald[14], Jeffrey P. Gorski[15], Mohsen Ghanbari[3,16], Serkalem Demissie[17], Gustavo Duque[18], Matthew T. Maurano[19], Douglas P. Kiel[20,21,22], Yi-Hsiang Hsu[20,22,23], Bram C.J. van der Eerden[1], Cheryl Ackert-Bicknell[24], Sjur Reppe[25,26], Kaare M. Gautvik[26,27], Truls Raastad[28], David Karasik[20,29], Jeroen van de Peppel[1], Vincent W.V. Jaddoe[2], André G. Uitterlinden[1,2,3], Jonathan H. Tobias[30], Struan F.A. Grant[9,11,31], Pantelis G. Bagos[6], David M. Evans[4,5] & Fernando Rivadeneira [1,2,3]

Bone mineral density is known to be a heritable, polygenic trait whereas genetic variants contributing to lean mass variation remain largely unknown. We estimated the shared SNP heritability and performed a bivariate GWAS meta-analysis of total-body lean mass (TB-LM) and total-body less head bone mineral density (TBLH-BMD) regions in 10,414 children. The estimated SNP heritability is 43% (95% CI: 34–52%) for TBLH-BMD, and 39% (95% CI: 30–48%) for TB-LM, with a shared genetic component of 43% (95% CI: 29–56%). We identify variants with pleiotropic effects in eight loci, including seven established bone mineral density loci: *WNT4*, *GALNT3*, *MEPE*, *CPED1/WNT16*, *TNFSF11*, *RIN3*, and *PPP6R3/LRP5*. Variants in the *TOM1L2/SREBF1* locus exert opposing effects TB-LM and TBLH-BMD, and have a stronger association with the former trait. We show that *SREBF1* is expressed in murine and human osteoblasts, as well as in human muscle tissue. This is the first bivariate GWAS meta-analysis to demonstrate genetic factors with pleiotropic effects on bone mineral density and lean mass.

[1] Department of Internal Medicine, Erasmus MC University, Rotterdam 3015GE, The Netherlands. [2] The Generation R Study Group, Erasmus Medical Center, Rotterdam 3015GE, The Netherlands. [3] Department of Epidemiology, Erasmus Medical Center, Rotterdam 3015GE, The Netherlands. [4] University of Queensland Diamantina Institute, Translational Research Institute, Brisbane, Queensland 4102, Australia. [5] MRC Integrative Epidemiology Unit, University of Bristol, Bristol BS8 2BN, UK. [6] Department of Computer Science and Biomedical Informatics of the University of Thessaly, Lamia GR 35100, Greece. [7] Department of Hygiene and Epidemiology, University of Ioannina Medical School, Ioannina 45110, Greece. [8] COPSAC, Copenhagen Prospective Studies on Asthma in Childhood, Herlev and Gentofte Hospital, University of Copenhagen, Copenhagen 2820, Denmark. [9] Division of Human Genetics, Children's Hospital of Philadelphia, Philadelphia, PA 19104, USA. [10] Division of GI, Hepatology, and Nutrition, Children's Hospital of Philadelphia, Philadelphia, PA 19104, USA. [11] Department of Pediatrics, Perelman School of Medicine, University of Pennsylvania, Philadelphia, PA 19104, USA. [12] Steno Diabetes Center Copenhagen, Gentofte 2820, Denmark. [13] Department of Epidemiology and Biostatistics, School of Public Health, Imperial College London, London W2 1PG, UK. [14] Departments of Anatomy and Cell Biology and Orthopaedic Surgery, School of Medicine, Indiana University, Indianapolis, IN 46202, USA. [15] Department of Oral and Craniofacial Sciences, School of Dentistry, University of Missouri-Kansas City, Kansas City, MO 64108, USA. [16] Department of Genetics, School of Medicine, Mashhad University of Medical Sciences, Mashhad, Iran. [17] Department of Biostatistics, Boston University School of Public Health, Boston, MA 02131, USA. [18] Australian Institute for Musculoskeletal Science (AIMSS), The University of Melbourne and Western Health, St Albans, Victoria 3021, Australia. [19] Institute for Systems Genetics, New York University Langone Medical Center, New York, NY 10016, USA. [20] Hebrew SeniorLife, Institute for Aging Research, Roslindale, MA 02131, USA. [21] Department of Medicine Beth Israel Deaconess Medical Center and Harvard Medical School, Boston, MA 02215, USA. [22] Broad Institute of MIT and Harvard, Boston, MA 02115, USA. [23] Molecular and Integrative Physiological Sciences, Harvard School of Public Health, Boston, MA 02115, USA. [24] Center for Musculoskeletal Research, University of Rochester, Rochester, NY 14642, USA. [25] Department of Medical Biochemistry, Oslo University Hospital, Ullevaal, 0450 Oslo, Norway. [26] Unger-Vetlesen Institute, Oslo Diakonale Hospital, 0456 Oslo, Norway. [27] Department of Molecular Medicine, University of Oslo, 0372 Oslo, Norway. [28] Department of Physical Performance, Norwegian School of Sports Sciences, 0863 Oslo, Norway. [29] Faculty of Medicine in the Galilee, Bar-Ilan University, Safed 1311502, Israel. [30] School of Clinical Sciences, University of Bristol, Bristol, BS10 5NB, UK. [31] Division of Endocrinology, The Children's Hospital of Philadelphia, Philadelphia, PA 19104, USA. Carolina Medina-Gomez and John P. Kemp contributed equally to this work. David M. Evans and Fernando Rivadeneira jointly supervised this work. Correspondence and requests for materials should be addressed to F.R. (email: f.rivadeneira@erasmusmc.nl)

The interaction between skeletal muscle and bone has long been perceived as being mechanical, where bone provides an attachment site for muscles and muscles apply forces to bone. Such muscle-derived forces drive an adaptive response in bone affecting its morphology, structure, and strength as portrayed by the mechanostat theory[1]. Whenever mechanical stimuli exceed a set point, bone tissue is added at the location, where it is mechanically necessary. As muscle mass is a key determinant of bone mineral density (BMD) variation, it is expected that these traits are phenotypically highly correlated[2–4]. Recent studies show that the coupling between these two tissues is much more complex[5, 6]. It extends beyond the well-established mechanical interaction, already beginning during embryonic development, when osteoblasts and muscle cells share a common mesenchymal precursor[6]. Indeed, there is constant paracrine crosstalk between bone and muscle throughout life, with both sharing common responses to paracrine and endocrine stimulation[7]. Under this perspective of coupled development and physiological relationships during growth, it is highly likely that both tissues share genetic determinants exerting pleiotropic effects[8].

BMD, measured by dual-energy X-ray absorptiometry (DXA), is commonly used in clinical practice to assess bone health in young populations and to diagnose osteoporosis and determine fracture risk in the older populations. Conveniently, lean mass, which is a good proxy for skeletal muscle mass[9, 10], can also be derived from the same whole-body DXA scans. A previous study in postmenopausal women[9] showed DXA lean mass has a high correlation ($\rho$=0.94) with skeletal muscle mass of the whole body measured by magnetic resonance imaging (MRI). Furthermore, another study in peri-pubertal children[10] showed a 0.98 correlation between DXA derived lean mass in the mid third femur and skeletal muscle mass of the same region measured with MRI.

Heritability studies have demonstrated that between 50 and 85% of BMD variation can be explained by genetic factors[11], whereas heritability estimates for muscle phenotypes, such as lean mass, grip strength, arm flexion, and leg strength, range between 30 and 65%[12, 13]. Genome-wide association studies (GWAS) have been successful in revealing the genetic architecture of BMD, identifying >60 different loci robustly associated with the trait at different skeletal sites[14–26]. In contrast, very few robust associations have been reported to date by GWAS of lean mass[27]. Notwithstanding, twin studies have calculated the additive genetic correlation of BMD and lean mass to range from 30 to 45%[28].

GWAS usually evaluate one trait at a time, however, recent methods enable multivariate analyses to be performed. Multivariate methodologies offer several advantages over univariate GWAS including, increased power (when traits are genetically correlated), reduction of the multiple testing burden and enhanced biological insight (in the case of pleiotropy)[29]. Therefore, this approach is well-suited to study the genetic influence over the bone-muscle unit. We applied a bivariate GWAS approach for total-body lean mass (TB-LM) and BMD less head region (TBLH-BMD) in four pediatric cohorts leading to the identification of variants in eight different loci with statistical evidence for pleiotropic effects on both traits. Seven of these are established BMD loci, while the 17p11.2 locus is reported for the first time as associated with musculoskeletal traits. Our functional follow-up points to SREBF1 as the most likely gene underlying the association signal. SREBP1, the product of SREBF1 is implicated in osteoblast and myoblast differentiation by previous studies. Our findings shed a light into the role of SREBF1 in the crosstalk between bone and muscle during development.

## Results

**SNP heritability and genetic correlation.** Our study includes data from 10,414 participants from four different pediatric cohorts (Table 1, Supplementary Note). In all four studies TBLH-BMD and TB-LM phenotypic correlation ($\rho$) adjusted for sex, age, and fat percent was significant ($P < 0.01$) and of similar magnitude, Generation R ($\rho$=0.44), ALSPAC ($\rho$=0.45), BMD-CS ($\rho$=0.49), and COPSAC ($\rho$=0.42). GCTA calculated SNP heritability and genetic correlation for adjusted TBLH-BMD and TB-LM were estimated for the Generation R ($N = 3027$) and ALSPAC ($N = 4820$) studies, where there was sufficient statistical power for the analysis (Table 2). Significant SNP heritability estimates ranged between 30 and 45% for both traits in these two cohorts, with a genetic correlation of ~ 30% in both studies. Similar results were derived using LD-score methodology, which estimated the heritability of TBLH-BMD to be 43% (CI: 34–52%), 39% (CI: 30–48%) for TB-LM and a shared genetic component of 43% (CI: 29–56%).

**Univariate and bivariate GWAS of lean mass and BMD.** Univariate GWAS meta-analysis identified variants associated at the genome-wide significant (GWS, $5\times10^{-8}$) level with TBLH-BMD mapping to four different loci all of which have been previously associated with BMD in adult and/or children[15, 17, 19–22], shown in Fig. 1: the 7q31.31 WNT16/CPED1 locus, leading SNP rs917727-T (beta = 0.129 SD, $P = 1.28\times10^{-16}$); the 11q13.2 LRP5/PPP6R3 locus, leading SNP rs12272917-C (beta= − 0.097 SD, $P = 1.42\times10^{-9}$); the 1p36.12 WNT4 locus, leading SNP rs3765350-G (beta= − 0.094 SD, $P = 9.75\times10^{-9}$); and the 2q24.3 GALNT3 locus, leading SNP rs6726821-G (beta= − 0.077 SD, $P = 2.95\times10^{-8}$). A summary of all genome-wide associated SNPs can be found in Supplementary Data 1. The univariate GWAS meta-analysis of TB-LM yielded no GWS associations along the lines of the sparse number of identified loci in adults[27]. QQ-Plots for these two analyses showed no evidence for early inflation as consequence of

### Table 1 Anthropometric characteristics of study participants

| Mean (SD) | Generation R | | ALSPAC | | BMD-CS | | COPSAC | |
|---|---|---|---|---|---|---|---|---|
| | n = 4071 | | n = 5251 | | n = 821 | | n = 273 | |
| Age, years | 6.21 | 0.32 | 9.94 | 0.32 | 8.74 | 1.91 | 6.89 | 0.72 |
| Women (%) | 2035 | 49.98% | 2673 | 50.90% | 431 | 61.80% | 144 | 53.14% |
| CEU ancestry (%) | 2171 | 53.32% | 5251 | 100% | 634 | 77.22% | 271 | 100% |
| Height (m) | 1.19 | 0.63 | 1.4 | 0.64 | 1.32 | 0.12 | 1.24 | 0.61 |
| Weight (kg) | 23.08 | 4.08 | 34.7 | 7.41 | 30.78 | 8.76 | 24.6 | 4.25 |
| TBLH BMD (g cm$^{-2}$) | 0.555 | 0.05 | 0.777 | 0.053 | 0.67 | 0.10 | 0.583 | 0.05 |
| TBLH BMC (g) | 528.5 | 104.21 | 891.92 | 181.85 | 726.12 | 205.92 | 2932 | 591.04 |
| TB Lean Mass (g) | 16,393 | 2284 | 24,553 | 3184 | 22,360 | 5830 | 17,460 | 2600 |
| TB Fat Mass (g) | 5862 | 2328 | 8561 | 5108 | 7590 | 3580 | 7010 | 2490 |

**Table 2 Trait SNP heritability and genetic correlation TBLH-BMD/TB-LM in pediatric populations**

| Study | N | TBLH-BMD Heritability | | | TB-LM Heritability | | | Genetic correlation TBLH-BMD/TB-LM | | |
|---|---|---|---|---|---|---|---|---|---|---|
| | | $h^2$ | SE | P | $h^2$ | SE | P | ρ | SE | P |
| Generation R | 3028 | 0.311 | 0.12 | 0.004 | 0.400 | 0.12 | $3\times10^{-4}$ | 0.299 | 0.21 | 0.126 |
| ALSPAC | 4820 | 0.437 | 0.07 | $6\times10^{-10}$ | 0.325 | 0.07 | $3\times10^{-6}$ | 0.323 | 0.12 | 0.016 |

Only unrelated individuals (e.g., no two individuals in the analysis were closer than third degree cousins in either of the two studies) were included in the analysis

bias (i.e., population stratification, cryptic family relatedness or genotyping errors) (Supplementary Fig. 1).

On the basis of the covariate adjusted TBLH-BMD/TB-LM bivariate analysis, the genomic inflation factor λ was 1.08. In contrast to the two univariate analyses, the bivariate meta-analysis identified eight different GWS signals (Fig. 1). Four of them were not found to be GWS in the TBLH-BMD analysis mapping to the 4q22.1 *MEPE* locus; the 13q14.11 *TNFSF11* locus; the 14q2.12 *RIN3* locus, all known BMD loci, and the 17p11.2 *TOM1L2/SREBF1* locus (Table 3, Supplementary Fig. 2). Signals in the 2q24.3, 11q13.2, and 17p11.2 loci presented nominal evidence of association ($P < 0.05$) with both traits (Table 3, Supplementary Data 2). Detailed description of all these known BMD loci can be found in the Supplementary Note.

The signal on 17p11.2 (lead SNP rs7501812, $P = 1.4\times10^{-10}$) was the only GWS signal where association was stronger with TB-LM than with TBLH-BMD, mapping to a region not previously associated with neither trait. The region underlying this signal (Fig. 2) extends along an LD-block harbouring several genes including *MYO15A*, *LRRC48*, *MIR33B*, *C17orf39 [GID4]*, *DRG2*, *RAI1*, *SREBF1*, *TOM1L2*, *ATPAF2*, the latter seven all shown to be expressed in skeletal muscle[30]. Detailed information for the genes residing at 17p11.2 is provided in Supplementary Table 1. All GWS SNPs in this region yielded nominally significant opposite effect for the coded allele in TBLH-BMD as compared to TB-LM, despite the positive correlation between these traits.

Although using different phenotypes than the ones used in our analysis, Genetic bivariate strategies in adult populations have approached the bone/lean mass relationship. A GWAS bivariate analysis of bone size and appendicular lean mass in Chinese and European individuals[31] reported a potential association signal mapping to the *GLYAT* gene (11q12.1) arising from a low frequency (MAF < 0.05) variant not present in our meta-analysis. In addition, a linkage study reported a significant signal (LOD score = 4.86) mapping to the 15q13 locus and multiple suggestive signals (LOD score < 3) in 7p22, 7q21, 7q32, and 13q11[32]. In our bivariate GWAS meta-analysis none of these regions (including *GLYAT*) contained significantly associated SNPs ($P < 9.0\times10^{-6}$ after multiple correction, 5537 SNPs tested). Additionally, we applied a bivariate analysis to the summary statistics of previously reported univariate GWAS meta-analyses of BMD and lean mass traits in adults[20, 27] (Supplementary Data 3). We only found evidence of genetic variants exerting pleiotropic effects on both traits in the 11q13.2 locus. However, consistent with our findings in children, variants in the 17p11.2 locus (led by rs7501812) showed opposite association with lumbar spine BMD and TB-LM at the margin of reaching statistical significance ($P < 0.07$).

**Gene annotation and eQTL analyses of the 17p11.2 locus**. We identified 163 proxy SNPs ($r^2 > 0.8$) of our GWS SNPs in the 17p11.2 region, of which 78 were present in our meta-analysis (Supplementary Table 5). Only rs11868035 (in high LD with a GWS SNP (rs11654081, $r^2 > 0.8$)), was previously reported in the GWAS catalog. The G-allele of this SNP has a protective effect on

Parkinson's disease as reported in a recent meta-analysis comprising ~ 30,000 individuals ($P = 5.6\times10^{-8}$)[33]. The rs11868035 G-allele was nominally associated with lean mass (beta=0.05SD; $P = 3\times10^{-4}$) and approached GWS in our bivariate meta-analysis ($P_{bivariate}=8\times10^{-7}$). In the proximity of these variants and in high LD ($r^2 > 0.8$) with rs11654081, we identified two *RAI1* common missense variants with no clinical annotation (Supplementary Data 4). The rs11649804/Pro165Thr variant ($P_{bivariate}=9\times10^{-7}$) was predicted to be 'tolerated' by SIFT and 'probably damaging' by PolyPhen-2, while the rs3803763/Gly90Ala variant was not present in our meta-analysis.

We identified two miRNAs, miR-6777 and miR-33b, in the associated region. Of these, miR-33b is located in an intronic region of the *SREBF1* gene and is co-transcribed with its host gene. Furthermore, analysis of the miR-33b putative targets disclosed this miRNA as a potential regulator of both myogenesis and osteogenesis by targeting multiple genes involved in the related pathways, such as *TPM3* and *BMP3*[34, 35]. Moreover, we found that 5 GWS markers in the 3′UTR of *TOM1L2* are located in putative miRNA-binding sites (i.e., rs3183702, rs9915248, rs3744115, rs1052299 and rs1108648). Of these, rs1052299 was predicted to create novel-binding sites for two highly conserved miRNAs affecting bone and muscle[36–38] -miR-133a-5p and miR-138-5p—and is more likely to affect miRNA-mediated regulation of *TOM1L2*.

As most of the association signal in 17p11.2 arises from non-coding variants, we reviewed the possible regulatory annotation of these SNPs by using data from the ENCODE and ROADMAP EPIGENOMICS projects through the UCSC browser (Fig. 3). Shadowed areas correspond to CTCF (CCCTC-binding factor)-associated areas of chromatin interaction by looping. As these experiments are not available in musculoskeletal cells, we examined interactions in MCF-7 and K562 cells, for which the CTCF binding sites coincide with those predicted to exist in the musculoskeletal cells. Three selected interacting regions are shown: the first one brings together an intronic region of *TOM1L2* and the 3′region of *SREBF1;* the second one, contains two regions within *TOM1L2;* and the third one, intronic regions from *TOM1L2* and *ATPAF2*. The topological associated domains (TADs) in the region are in line with these results, evidencing areas of complex chromatin structures (Supplementary Fig. 3).

The SNP with the strongest bivariate association with TBLH-BMD and TB-LM in this locus, rs7501812, is also a *cis*-eQTL variant found to regulate the expression of *SREBF1*, *C17orf39 [GID4]*, *TOM1L2* and *ATAPF2* (FDR < 0.05) in whole blood, based on expression data from 5311 non-transformed peripheral blood samples and publicly available in the Blood eQTL browser[39]. Nonetheless, *SREBF1* expression, represented by two probes in the data set, showed the highest correlation with rs7501812. The G-allele, associated with higher TBLH-BMD (beta = 0.043 SD; $P = 2.0\times10^{-3}$) and lower TB-LM (beta= − 0.056 SD; $P = 5.5\times10^{-5}$) in our study, decreased the expression of the two *SREBF1* probes in whole blood. By interrogating the browser per gene, rather than per SNP, we generally found that alleles from the bivariate GWS SNPs associated with higher TBLH-BMD

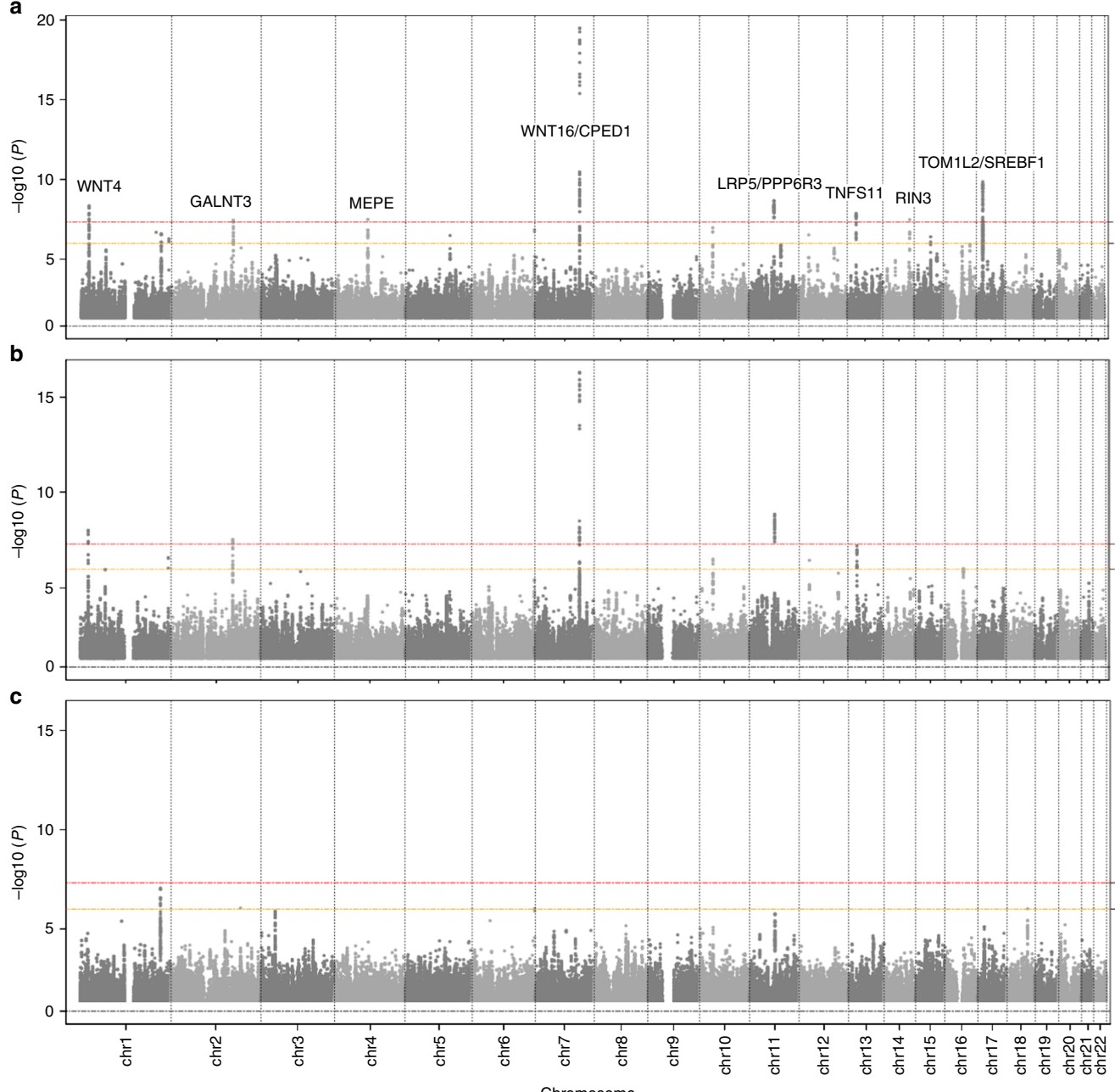

**Fig. 1** Manhattan plot of the meta-analyses for musculoskeleta traits. **a** TBLH-BMD and TB-LM bivariate meta-analysis. **b** TBLH-BMD meta-analysis. **c** TB-LM meta-analysis. *Dashed red and yellow lines* mark the GWS threshold ($P < 5 \times 10^{-8}$) and suggestive threshold ($P < 1 \times 10^{-6}$), respectively

and lower TB-LM in the region, are associated with decreased expression of *SREBF1*, *TOM1L2*, and *C17orf39 [GID4]* but increased expression of *ATPAF2* in whole blood (Supplementary Data 4). We applied the Summary Mendelian Randomization (SMR) approach[40] to prioritize genes underlying the GWS signal in the 17p12 region. Results from this approach combining Mendelian randomization and heterogeneity in dependent instruments (HEIDI) tests using the Blood eQTL browser data set[39], indicated that *SREBF1* is the most likely gene driving the associations with TB-LM ($P_{SMR}=1.2\times10^{-4}$; $P_{HEIDI}=0.23$) and TBLH-BMD ($P_{SMR}=2.9\times10^{-3}$; $P_{HEIDI}=0.79$) (Supplementary Fig. 4). Expression data from GTEx also showed a highly negative significant association of rs7501812-G allele with the expression of *SREBF1* in whole blood ($P=2.4\times10^{-14}$). In contrast, GTEx data in skeletal muscle showed a positive correlation for

rs7501812-G with *SREBF1* expression ($P=6.2\times10^{-4}$). Furthermore, rs4925114 ($P=3.5\times10^{-6}$) and rs11654081 ($P=6.0\times10^{-6}$) major alleles (LD ~ 0.75 with rs7501812-G), significantly upregulated the expression of *SREBF1* in skeletal muscle (FDR<0.05) (Supplementary Fig. 5).

**Functional characterization of *SREBF1*.** Leveraging mouse calvaria-derived cells, we assessed gene expression during osteoblastogenesis, which revealed that *Rai1*, *Tom1l2*, *Atpf2*, and *Srebf1* were expressed in this cell type without major changes from preosteoblast to its mature stage (Supplementary Fig. 6). In contrast, *Lrrc48* was not expressed at all in the pre- or mature osteoblast, as determined by RNAseq profiles. We also assessed expression profiles during human mesenchymal stem cell (hMSC)

**Table 3 Lead SNPs for the eight bivariate association signal with TBLH-BMD and TB-LM**

| Locus | SNP | CHR | Position | A1 | A2 | EAF | Beta TBLH-BMD | P TBLH-BMD | Beta TBLM | P TBLM | P bivariate |
|---|---|---|---|---|---|---|---|---|---|---|---|
| 1p36.12 | rs6684375 | 1 | 22579021 | T | C | 0.19 | 0.0842 | 2.47E-06 | −0.0273 | 0.14 | 4.69E-09 |
| 2q24.3 | rs6726821 | 2 | 166286360 | G | T | 0.45 | −0.0769 | 2.95E-08 | −0.0104 | 0.48 | 3.56E-08 |
| 4q22.1 | rs7672749 | 4 | 89017308 | A | G | 0.09 | 0.0822 | 4.35E-04 | −0.0591 | 0.01 | 3.29E-08 |
| 7q31.31 | rs917727 | 7 | 120805815 | T | C | 0.29 | 0.1289 | 5.29E-17 | −0.0042 | 0.77 | 3.07E-20 |
| 11q13.2 | rs12284933 | 11 | 68076065 | A | G | 0.24 | −0.0969 | 1.85E-09 | −0.0699 | 1.49E-05 | 2.19E-09 |
| 13q14.11 | rs9525638 | 13 | 42026577 | C | T | 0.42 | 0.0756 | 6.42E-08 | −0.0006 | 0.97 | 1.35E-08 |
| 14q2.12 | rs754388 | 14 | 92185163 | G | C | 0.18 | −0.0862 | 3.26E-06 | 0.0210 | 0.24 | 3.36E-08 |
| 17p11.2 | rs7501812 | 17 | 17691632 | G | A | 0.41 | 0.0431 | 0.002 | −0.0563 | 5.53E-05 | 1.44E-10 |

Estimates were derived from 10,414 children participants of four different pediatric studies worldwide. Beta coefficients and allele frequencies (EAF) are reported for the A1 allele

differentiation using qPCR data. The expression of *SREBF1* relative to *GAPDH* in two donors, showed that *SREBF1* expression peaks at the onset of osteoblast mineralization. In contrast, the expression of *TOM1L2* was not detected above background level (Fig. 4). *SREBF1* is an adipocyte differentiation factor (ADD-1) that produces SREBP-1, a transcription factor ubiquitously expressed (more strongly in lipogenic tissues) and directly regulating the transcription of over 200 genes involved in the de novo synthesis of fatty acids, triglycerides, and cholesterol[41]. SREBP-1, in its active form, is important for the mineralization of osteoblastic cultures in vitro, as its overexpression increases the number of mineralized foci[42]. Contrary to its positive regulatory role in osteoblast differentiation and mineralization, in skeletal muscle SREBP-1 protein indirectly downregulates *MYOD1*, *MYOG*, and *MEF2C*, acting as a key regulator of myogenesis. Similarly, overexpression of SREBP-1 inhibits myoblast-to-myotube differentiation [43], reduces cell size and leads to loss of muscle-specific proteins in differentiated myotubes[41].

We evaluated expression profiles of the genes showing putative interactions from the ENCODE/Roadmap Epigenomics analysis, for association with bone and muscle phenotypes. *SREBF1* muscle expression, assessed from postmenopausal women thigh muscle biopsies ($N = 18$) showed significant negative correlation with femoral neck BMD of the donor ($P < 0.001$) and was borderline significantly associated with TBLH-BMD ($P = 0.05$) (Supplementary Table 2). Expression of *RAI1* was inversely correlated with thickness of the *vastus lateralis* muscle of the donor ($P = 0.01$), while *TOM1L2* expression levels were positively correlated with this trait ($P = 0.02$). *ATPAF2* and *C17orf39* [*GID4*] did not correlate with any of the scrutinized measurements of the donors. Evaluation of expression profiles from trans-Iliac bone biopsies in a separate group of postmenopausal women ($N = 80$), revealed no correlation ($P < 0.05$) with donor phenotypes for either of the genes despite being expressed in bone. To note, SREBP-1 has also been shown to interact with lamin A, implicated in muscular dystrophy[44], and this mechanism cannot be ruled out as key in the association of this gene with muscle outcomes. In addition, miR-33b may function in concert with the SREBP-1 host gene product to regulate myogenesis or/and osteogenic differentiation, as it does in controlling lipid homeostasis[45].

## Discussion

This work illustrates the enhanced power of bivariate analysis to identify associations not detected by the univariate analysis of correlated traits, and to hint at pleiotropic effects. This bivariate GWAS meta-analysis of bone mineral density and lean mass in children identified eight loci associated with both traits. Seven of the identified loci have been reported as associated with BMD in previous GWAS of adult and paediatric populations. The 17p11.2 is a novel locus, not previously associated with lean mass or BMD,

marked by a long stretch of LD harbouring multiple genes in the region including *RAI1*, *LRRC48*, *MIR33B*, *C17orf39*, *DRG2*, *MYO15A*, *SREBF1*, *TOM1L2*, and *ATPAF2*. Different lines of evidence, arising from in-silico and in-vivo functional follow-up suggest that *TOM1L2* or *SREBF1* are the main candidate genes underlying the bivariate association signal; confirming the existence of pleiotropic effects on BMD and lean mass arising from the 17p11.2 region.

SNP heritability estimated both by GCTA and LD score regression confirm that TBLH-BMD and TB-LM are moderately high heritable traits. In addition, we found that more than one-third of the tagged additive genetic effect is shared between TBLH-BMD and TB-LM. These estimates are lower than those derived from previous twin studies one of them estimating the heritability for TBLH-BMD to be between 80 and 90%, between 70 and 88% for TB-LM and with a genetic correlation (shared heritability) of 46%[28]. Similar to other complex traits, SNP heritability estimates for TBLH-BMD and TB-LM were lower than heritability estimates derived from family studies[46, 47]. This could be due to inflation of the family based heritability estimates[46, 47] but also a consequence of the amount, and type, of markers being surveyed, limited to the genotyped SNPs and the underlying variants that they tag. Thus, if the selected markers are not sufficiently correlated (LD tagging) with the true genetic variants explaining the trait-variance, SNP heritability would certainly be less than true heritability.

The univariate approach for TBLH-BMD identified four GWS signals, all of them reported before in BMD GWAS at different skeletal sites[19–21]. However, we did not find any GWS result for TB-LM in our meta-analysis in pediatric populations. This is in line with the scarce number of loci identified with lean mass variation in adults. To date, only five loci have been identified as associated with TB-LM in an study comprising > 80,000 adults[27]. It is worth noting that lean mass in children has a higher proportion of organ tissue vs. skeletal mass as compared to adults. This could hamper the identification of loci influencing muscle mass and claims for new larger efforts in children. Yet, implementing a bivariate meta-analytical approach we identified eight associated loci, three of which were nominally associated ($P < 0.05$) with both TBLH-BMD and TB-LM. Therefore, the bivariate analysis is an alternative approach to diminish such power limitations. Our bivariate approach showed particularly high statistical power to identify variants, in which the coded allele exerted an opposite effect on TBLH-BMD as compared to TB-LM. Our meta-analysis did not replicate potential pleiotropic signals previously reported in adults[31, 32]. Nevertheless, the bivariate analysis of summary statistics of published BMD and lean mass efforts in adults we did, identified variants with strong evidence for pleiotropic effects in the 11q13.2 locus. The lack of replication in the other regions may well be a consequence of differential genetic effects during the life course, e.g., the

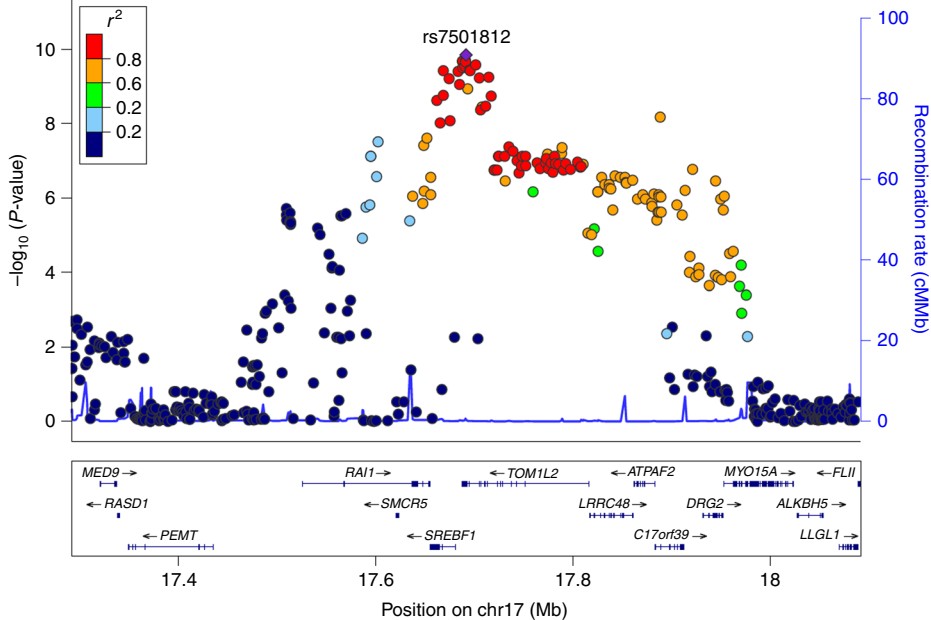

**Fig. 2** Regional association plot for the bivariate meta-analysis of TBLH-BMD and TB-LM displaying the 17p11.2 locus. Genetic coordinates are as per Hapmap phase II-CEU

accumulation of environmental factors attenuating effects in the elderly. Overall, the role of protein products of genes such as *LRP5*, *TNSF11*, *WNT16*, *MEPE*, and *WNT4* -mapping to the identified loci- in bone metabolism is long-established. However, none of the genes in the identified loci has been yet implicated in muscle metabolism and opens up new opportunities to study pleiotropic effects and the interplay between bone and muscle.

We further focused on the 17p11.2 locus, which is novel in the spectrum of musculoskeletal traits GWAS, and which is the only associated region where the signal was stronger for TB-LM than for TBLH-BMD. Despite the positive correlation between TB-LM and TBLH-BMD, variants in this region exerted opposite effects on these traits, probably facilitating their discovery in a bivariate analysis powerful setting. Moreover, this region was recently described as a QTL for BMD phenotypes in rats[48]. The authors attributed the association signal to *Rai1*, as knockout mice for this gene present with retarded growth and display malformations in both the craniofacial and the axial skeleton[49]. *RAI1* is responsible for the Smith-Magenis syndrome in humans, that as in mice presents with craniofacial and skeletal abnormalities[50]. Nonetheless, the two common missense variants in LD with the bivariate GWS SNPs in our meta-analysis, have not been related to the disease.

We retrieved additional lines of evidence to disentangle the implication of gene(s) underlying the association signal although results were not conclusive. The regulatory landscape across the long stretch of LD in this region contains complex chromatin interactions that are consistent with the potential involvement of more than one gene (i.e., *SREBF1*, *TOM1L2*, and *ATPAF2*). TADs were retrieved for two oncogenic cell lines rather than for osteoblasts and skeletal muscle cells. However, TADs have been shown to be largely invariant across cell types, and physiological conditions[51]. Therefore, we believe similar interactions might also be taking place in these cells, suggesting that associated SNPs may be influencing distant genes within the TAD. For example, in agreement with the predicted complex chromatin looping in this locus, the lead SNP of the bivariate association signal (rs7501812), has strong correlation with the expression of various genes in the

region as seen in whole blood. However, its correlation with *SREBF1* is the strongest one. Also, SMR implicates *SREBF1* as the most likely candidate gene underlying the 17p11.2 signal. Yet, *TOM1L2* or *RAI1* probes were not included in the analysis as they lacked a GWS *cis*-eQTL. Such stringency in the SMR inclusion criteria results from the basic assumption of the instrument used in the Mendelian randomisation approach, i.e., the SNP should be strongly associated with exposure. Further, the e-QTL effects identified in whole blood may not constitute good proxies for eQTLs exerting tissue specific effects. However, there is no eQTL data available from other tissues (particularly bone and muscle) with comparable statistical power to the one provided by the large-scale eQTL study in whole blood used here.

In addition, rs4925114 and rs11654081, both in high LD with rs7501812, show the strongest correlation with the expression of *SREBF1* in skeletal muscle. Interestingly, the same allele in either of these three SNPs correlates with lower *SREBF1* expression in whole blood and higher *SREBF1* expression in skeletal muscle. Unfortunately, the GTEx eQTL database has no data available for bone related cells. Our ex-vivo models in murine osteoblasts established that *Rai1*, *Atpaf2*, *Srebf1*, and *Tom1l2* were expressed in bone. With the exception of *Srebf1*, knockout mice for these genes all present with a skeletal phenotype. While *Srebf1* KO models have been reported in the literature, we have no certainty that bone phenotyping was performed on these mice.

We also confirmed the expression of *SREBF1* in human MSC-derived osteoblasts. Further, analysis of expression profiles in muscle biopsies from post-menopausal women showed a correlation between the expression of *RAI1* and *TOM1L2* and muscle thickness, and noteworthy, a negative correlation between *SREBF1* expression and bone density parameters. Altogether, *SREBF1* and *TOM1L2* are the strongest candidates underlying the genetic association signal. The role of active SREBP-1, *SREBF1* product, exerting opposing effects on osteoblast and myoblast differentiation has been well documented[41–43]. However, despite the fact that the potential role of TOM1L2 in the musculoskeletal system remains to be elucidated, miRNAs predicted to regulate its expression have been shown to be involved in osteogenic

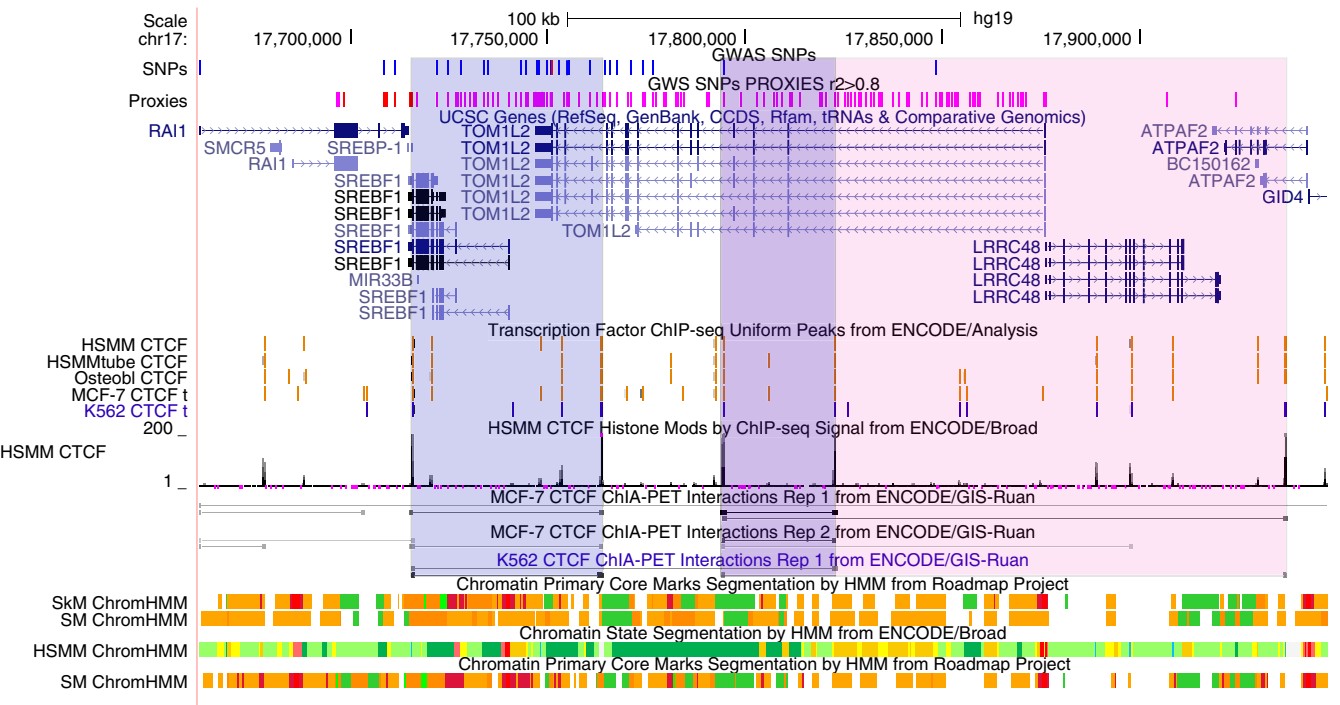

**Fig. 3** 17p11.2 locus displaying binding peaks and interaction tracks by CTCF ChIA-PET data together with histone marks based on chromatin state characterization. Predicted strong interactions shared in MCF-7 or K562 cell lines are shadowed in *blue* while a weaker interaction in both cell lines is shadowed in *pink*. These interactions overlap with predicted CTCF binding sites in osteoblast, myoblast and myotubes, hypothetically localizing SNPs in *TOM1L2* close to transcribed regions in *SREBF1* and A*TPAF2*. Histone markers in muscular cells display high representation of associated SNPs in active enhancer predicted regions. Chromatin states are defined as follows, *Bright Red*: Active Promote, *Orange*: Strong enhancer, *Yellow*: Weak/poised enhancer, *Blue*: Insulator, *Dark Green*: Transcriptional elongation, *Light Green*: Weak transcribed

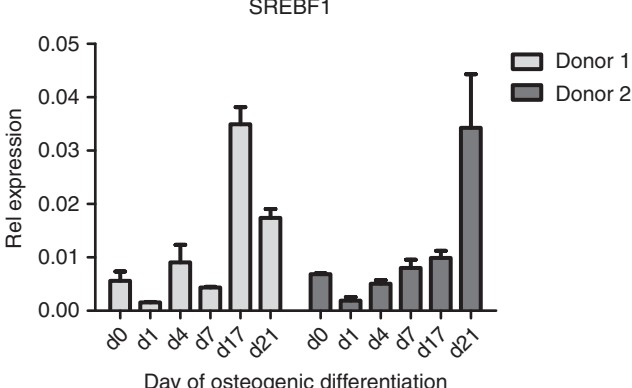

**Fig. 4** Expression profile of *SREBF1* during osteoblastogenesis. qPCR data of the *SREBF1* expression relative to GAPDH in two different donors. Error bars are defined as SD from two technical replicates at each time point. *SREBF1* expression peaks at onset of mineralization

differentiation and skeletal muscle development[36, 38]. In addition, the knockout mice of this gene observed with kyphosis[52].

Overall, there is compelling evidence that SREBF1 exerts opposite effects on the differentiation of myocytes and osteoblasts. These opposite effects are paralleled in the SNPs associated with TBLH-BMD/TB-LM in our bivariate approach, and therefore, it is likely that SREBF1 is underlying the association signal. The G-allele from rs7501812, shown to downregulate the expression of *SREBF1* in whole blood and upregulate it in skeletal muscle, is actually associated with higher TBLH-BMD and lower TB-LM (Fig. 5). Whereas, the expression of *SREBF1* in muscle biopsies correlated with lower BMD parameters in a direction

opposite to what is predicted by the SREBP-1 function in bone and muscle metabolism, the number of expression profiles included in analysis is small; also, this unexpected correlation could arise from tissue-specific effects. The eQTL analysis in GTEx showed that *SREBF1* is ubiquitously expressed and suggests that SNPs in 17p11.2 exert apparent opposite effect in blood and muscle. In addition, there are cellular processes that cannot be captured by our analyses and can be the origin of some inconsistencies seen at the association level. For instance, SREBP-1 must be proteolytically released from a transmembrane site in the Golgi complex to become active; then this activated fragment needs to be transported to the nucleus in order to exert its effect[53].

In summary, in this first large-scale bivariate GWAS meta-analysis of TBLH-BMD and TB-LM in pediatric cohorts, we identified eight GWS loci (in/near *WNT4, GALNT3, MEPE, CPED1/WNT16, TNFSF11, RIN3, TOM1L2/SREBF1,* and *PPP6R3/LRP5*). Although for most of these loci the association with TBLH-BMD is the dominant association responsible for their significance, we report in the 17p11.2 locus, a stronger association with TB-LM compared to TBLH-BMD. *SREBF1* arises as a compelling candidate to underlie this association, as this transcription factor is known to play an opposite role in osteoblast and myoblast differentiation; in line with the finding derived through our bivariate approach. Nevertheless, the level of evidence gathered does not allow us to discard fully that other genes in the region, particularly *TOM1L2*, may be implicated in these processes, and additional functional assessments are required. Alternatively, this study demonstrates that a considerable increase in power can be achieved when using a bivariate methodology as compared to testing each component of a multivariate phenotype individually, particularly in those cases where the effect of the associated variant goes in opposite directions in the involved

**Fig. 5** Schematic representation of the plausible role of rs7501812 in the pleiotropic modulation of BMD and lean mass. The G-allele from rs7501812 upregulates the expression of *SREBF1* both in skeletal muscle and bone. This overexpression would be expected to result in higher levels of the active form of SREBP-1. SREBP-1 exerts opposite effects on bone and muscle biogenesis. While it promotes osteoblast mineralization[45], it inhibits myoblast differentiation[44, 46]. Ultimately, this modulation would result in higher BMD and lower lean mass, as we observed in our bivariate GWAS analysis

phenotypes. Moreover, we show that this methodology may help discover previously unknown mechanisms of pleiotropic effects of genetic variants.

## Methods

**Study populations**. TBLH-BMD and TB-LM were measured in four pediatric population-based studies: the Generation R Study, the Avon Longitudinal Study of Parents and their Children (ALSPAC), the Bone Mineral Density in Childhood Study (BMDCS), and the Copenhagen Prospective Studies on Asthma in Childhood (COPSAC) cohort (Supplementary Note). All participants underwent TB-DXA scans; measurements were conducted by well-trained research assistants and daily quality control assurance was performed. Before the scan procedure, participants were asked to take off their shoes, heavy clothes, and metal accessories. As recommended by the International Society for Clinical Densitometry, TBLH-BMD was the measurement used in the analysis by all four studies. TB-LM and total body fat mass (TB-FM) were derived from the same scans. In addition, all individuals included in this study had genome-wide array data imputed to the HapMap Phase II reference panel (build 36 release 22) assessed as best guess genotypes, all markers with MAF < 0.05 were excluded from the analysis. The study was approved by the ALSPAC Law and Ethics committee (ALSPAC), The Ethics Committee for Copenhagen and the Danish Data Protection Agency (COPSAC), the Medical and Ethical Review Committee of the Erasmus University Medical Center (Generation R), and the institutional review boards of Children's Hospital of Los Angeles (Los Angeles, CA), Cincinnati Children's Hospital Medical Center (Cincinnati, OH), Creighton University (Omaha, NE), Children's Hospital of Philadelphia (Philadelphia, PA), and Columbia University (New York, NY) (BMDCS). Written informed consent was provided by all parents or custodians of the participants. A detailed description of each of these study populations is provided in Supplementary Note.

**Statistical methods**. TBLH-BMD and TB-LM measurements were adjusted by age, gender, height, fat percent (TB-FM/weight), and study specific covariates (genetic principal components and measurement center, when applicable). Standardized residuals were then generated. Phenotypic correlation ($\rho$) between these variables was evaluated by a Pearson correlation test in each study independently.

**SNP heritability and genetic correlation**. To characterize the extent to which common genetic variants determine pediatric BMD and lean mass, and the shared genetic etiology of these traits, we applied two recently proposed approaches for estimating SNP heritability and genetic correlation based on genome-wide sharing between distantly related individuals, one implemented in the GCTA software package[54, 55]. Precision of GCTA estimates is largely influenced by sample size and thus, analyses were performed considering genotyped markers only from the ALSPAC and Generation R studies. In the latter, given its multi-ethnic nature, relatedness coefficients were estimated with the Relatedness Estimation in Admixed Populations (REAP)[56] software. The obtained genetic relationship matrix (GRM) was then used as input for the GCTA analysis (Supplementary Methods). Samples were excluded from analysis to guarantee that no pair of individuals exceeded the standard GCTA cut-off coefficient of 0.025 for genetic relatedness (i.e., closer than third-degree cousins) in either of the two studies. In total 431 and 1043 individuals were excluded from ALSPAC and Generation R, respectively. The second approach implemented the LD-score regression methodology[57] to the TBLH-BMD and TB-LM meta-analyses results.

**Univariate and bivariate GWAS**. Analysis of the two quantitative traits was performed in each study individually based on best guess data using the "qt-command" available in PLINK[58]. This routine enables the user to obtain, in addition to the effect estimates, the means and standard deviations of both TBLH-BMD and TB-LM standardized residuals stratified by genotype. Meta-analysis of

2,276,811 SNPs, present in at least two of the studies, was performed using an inverse variance weighing. In the meta-analysis approach, we used a newly proposed method for bivariate meta-analysis, which is based on calculating the within-study covariances of the outcome specific estimates[59]. The algorithm uses a general approach and has been proposed for both, discrete and continuous outcomes, but in the case of continuous outcomes, such as the one encountered here, a direct extension of the method proposed by Wei and Higgins[60] is applied. We used the additive model of inheritance (per-allele mean difference of the quantitative phenotype) for both outcomes and the method of moments for estimation with the *mvmeta* command in Stata[61]. Source code and details are given in the webpage of the Department of Computer Science and Biomedical Informatics of the University of Thessaly, Greece. The overall bivariate association test was obtained by a standard Wald-type statistic (chi-square on 2 d.f.) that tests the null hypothesis that a particular SNP is not associated with either one of the outcomes. Genome-wide significance (GWS) was defined as a $P < 5 \times 10^{-8}$. Manhattan plots were generated in Easystrata[62].

**Adult bivariate analysis based on summary statistics of GWAS in musculoskeletal traits**. We assessed the evidence of association of the GWS markers identified here (149 SNPs, Supplementary Data 3), in a bivariate analysis of previous powered GWAS meta-analyses of lean mass (total body ($N \sim 38,000$))[27] and BMD (lumbar spine and femoral neck ($N \sim 32,000$))[20] performed in adult and elderly population. We chose to use a recently introduced method that performs bivariate GWAS allowing for mixed directions of effect known as empirical-weighted linear-combined test statistics (eLC), implemented in a $C^{++}$ eLX package and publicly available. Briefly, eLC directly combines correlated Z test statistics (calculated as β/SE derived from the univariate GWAS meta-analyses) with a weighted sum of univariate test statistics to empirically maximize the overall association signals and also to account for the phenotypical correlations the traits. In this case, we ran two analyses one incorporating the data from femoral neck BMD and total body lean mass meta-analyses, and another with data from lumbar spine and total body lean mass meta-analyses. Markers with nominal effect in both BMD and lean mass traits and with bivariate $P <= 3.35 \times 10^{-4}$ (0.05/149) were deemed to exert pleiotropic effects in adults.

**Gene Annotation and eQTL Analyses**. We identified all variants from the 1000 Genomes Project Phase 1 reference panel in strong linkage disequilibrium (LD) (CEU $r^2 > 0.8$) with the lead SNPs in the novel identified susceptibility loci, by bivariate analysis, using HaploReg3[63] for functional annotation. First, these proxy SNPs were scrutinized in the GWAS catalogue[64] to ascertain possible association with other traits. Second, we identified non-synonymous variants that were interrogated for likely downstream functional consequences using SIFT and PolyPhen-2 databases. Third, we interrogated two public databases, i.e., miR-NASNPv2[65] and PolymiRTSv3[66] for presence of any of the associated SNPs in miRNA genes or miRNA-binding sites within 3′UTRs of target genes. Other information about miRNAs, such as miRNA sequence, host gene, conservation and putative targets, were obtained from miRbase (release 21)[67] and TargetScan v7.1[68] websites. Fourth, variants were also evaluated for overlap with regions of predicted regulatory function generated by the Encyclopedia of DNA Elements (ENCODE) Project and the ROADMAP epigenomic project including: regions of enhancer activity, DNase I hypersensitivity, local histone modifications or proteins bound to these regulatory sites in cell lines of potential interest as HSMM (Skeletal Muscle Myoblast), HSMMtube (Skeletal Muscle myotubes differentiated from the HSMM), SKMC (skeletal muscle cells) and when present in the different assays, Osteobl (Osteoblasts). In addition, evidence of chromatin interactions was further explored by Chromatin Interaction Analysis by Paired-End Tag Sequencing (ChIA-PET) data from the ENCODE project in both MCF-7 and K562 oncologic cell lines. The UCSC Genome Browser was used for visualizing ENCODE data tracks indicative of regulatory function. Finally, topologically associated domains (TADs) for K562

cells containing the associated SNPs were retrieved from a recent Hi-C interactions browser[51].

The eQTL analysis was completed using recently published eQTL data sets from a meta-analysis of the transcriptional profiles from the peripheral blood cells of 5311 Europeans[39] and the GTEx database (GTEx v6p) based on 1641 samples across 43 tissues from 175 individuals[30].

**Summary data-based mendelian randomization.** We used the recently developed SMR approach[40] in order to identify the most likely gene in the lead SNP region, which has a complicated LD pattern. In short, the SMR approach uses the genetic variant as the instrumental variable to test for the causative effect of the gene expression (the exposure) on the phenotype of interest. This analysis is divided in two-steps. A transcriptome association analysis, where the effect of gene-expression on the trait of interest is expressed as a function of the GWAS summary statistics and the GWS eQTLs reported by Westra et al.[39] in whole blood. Then, a test for heterogeneity in dependent instruments (HEIDI) is applied; in principle, if the same causal underlying variant affects gene expression and the trait then, all SNPs in high LD with the causal variant will present the same SMR calculated effect. Therefore, testing for heterogeneity in the cis-eQTL region will be equivalent to testing if there is only one causal variant[40]. We used Bonferroni correction to account for multiple testing on 394 probes with GWS evidence for cis-eQTL effects in chromosome 17; this resulted in a significance $P$ threshold=$1.27 \times 10^{-4}$ (0.05/394). We used a conservative HEIDI threshold $P$=0.05, to exclude signals with high heterogeneity as used previously for different traits[40]. Plotting was performed in R including only the most significant probe per gene in the SMR analysis and using the lean mass GWAS results.

**Functional analyses of SREBF1.** Aiming to corroborate the functional or regulatory mechanism underlying the association between SREBF1 and LM and BMD, we used three different lines of evidence. The expression of SREBF1 was investigated in murine and human cells differentiating to osteoblasts. In addition, the expression of SREBF1 and its neighbouring genes TOM1L2 and ATPAF2 were examined in muscle biopsies from postmenopausal women. Methods specific to each analysis are described below.

**Murine pre-osteoblasts.** All procedures and use of mice for the neonatal osteoblast expression studies were approved by the Jackson Laboratory Animal Care and Use Committee (ACUC), in accordance with NIH guidelines for the care and use of laboratory animals. Pre-osteoblast-like cells were isolated from neonatal calvaria from C57BL/6J mice expressing cyan florescent protein (CFP) and RNA was collected at nine time points post differentiation, every other day for 18 days, starting 2 days after the cells were first exposed to an osteoblast differentiation cocktail. mRNA profiles for triplicate samples for each time point were generated by Next Generation High throughput RNA sequencing (RNAseq), using an Illumina HiSeq 2000.

**Human mesenchymal stem cells.** Human bone marrow derived mesenchymal stem cells ((hMSC, Lonza Group Ltd., Basel, Switzerland) were seeded in 12-well plates ($5 \times 10^3$ cells per cm$^2$) and differentiated into osteoblasts (using α-Mem pH7.5, 10% heat inactivated foetal calf serum (FCS), 100 nM Dexamethasone and 10 mM β-glycerophosphate). As mentioned in the datasheet provided by the company, cells were authenticated by FACS analyses for the presence of surface markers CD105, CD166, CD29, and CD44 and the absence of CD14, CD34 and CD45. In addition, osteogenic, adipogenic and chondrogenic differentiation was shown by alizarin red S staining, oil-red-O staining and collagen II staining, respectively. The human MSCs were tested negative for mycoplasma, both by the company and in-house during the culture experiments described in this manuscript. Total RNA was isolated using Trizol (Life Technologies, Carlsbad, CA, USA) after 0, 1, 4, 7, 17, and 21 days of differentiation. cDNA synthesis and quantitative polymerase chain reaction (qPCR) was carried out in duplicate in osteogenic differentiating hMSC from two different donors by using the following primers: SREBF1-for AGCCCCACTTCATCAAGGC, SREBF1-rev CAGAGACCAGGGGACTGAGA, TOM1L2-for GGCATTAACAATTGCCAGGCT and TOM1L2-rev CACTTGTGACACCCTCCTCC.

**Gene expression in human thigh muscle and bone biopsies.** Prior to these studies, validation, and recommendation were obtained by the Norwegian Regional Ethical Committee (REK no:2010/2539 and 2008/253 REK sør-øst D), and all sampling and procedures were according to the Law of Biobanking in Norway. All women who volunteered received a full clinical examination including the DEXA and laboratory analyses, and those who participated after fulfilling the inclusion criteria took pride in contributing to accomplish the aim of the study. Written informed consent was provided by all volunteers.

In muscle, gene expression profiles were generated from human thigh muscle biopsies donated by healthy (T-score >−1) postmenopausal women ($n = 18$), from whom anthropometric measurements and DXA data were also available. Expression profiling was performed using an Affymetrix HG U133 2.0 plus array. The Affymetrix Cel files were imported into Partek Genomics Suite (Partek Inc., St Louis, MO, USA), and normalized using the RMA (Robust Multichip Average) algorithm. Gene expression patterns were further adjusted for batch effects and

differing synthesis times[69]. The correlation of gene expression profiles of transcripts in SREBF1 and contiguous genes RAI1, TOM1L2 and ATPF2, and DXA parameters as femoral neck BMD, TBLH-BMD, TB-LM, TB-FM, together with muscular thickness of the *vastus lateralis*, was assessed using Spearman correlation. All parameters were adjusted for age, height and fat percent, except for TB-FM which was adjusted only for age and height.

For the bone study, trans Iliac bone biopsies ($n = 84$) were collected from postmenopausal women without underlying diseases other than osteoporosis or receiving medication (past or present) possibly affecting bone remodeling or representing secondary causes of osteoporosis[70]. RNA was purified and analyzed using Affymetrix HG U133 2.0 plus arrays as described elsewhere[70]. Correlation analysis were carried as described for the human muscle biopsies above, for all variables except muscle thickness which was not available.

**URLs.** Department of Computer Science and Biomedical Informatics, of the University of Thessaly, Greece [mvmeta command in Stata], (http://www.compgen.org/tools/multiple-outcomes)

HaploReg3, http://www.broadinstitute.org/mammals/haploreg/haploreg_v3.php

GWAS catalogue, http://www.ebi.ac.uk/gwas/

SIFT, http://sift.jcvi.org/

PolyPhen-2, http://genetics.bwh.harvard.edu/pph2/

miRNASNP v2, http://www.bioguo.org/miRNASNP/

PolymiRTS v3, http://compbio.uthsc.edu/miRSNP/search.php

miRbase http://www.mirbase.org/

TargetScan v7.1, http://www.targetscan.org/vert_71/

UCSC Genome Browser, https://genome.ucsc.edu/

Interactive Hi-C Data Browser, http://promoter.bx.psu.edu/hi-c/view.php

GTEx, http://www.gtexportal.org/home/

Blood eQTL browser, http://genenetwork.nl/bloodeqtlbrowser

Empirical-weighted linear-combined test statistics (eLC), https://sites.google.com/site/multivariateyihsianghsu/

**Data availability.** Summary statistics of all meta-analyses are available at the GEnetic Factors for OSteoporosis Consortium (GEFOS) website (http://www.gefos.org/?q=content/data-release-2017). Expression data from bone biopsies is available at the European Bioinformatics Institute (EMBL-EBI) ArrayExpress repository, ID: E-MEXP-1618. All other data supporting the findings of this study are available from the corresponding author upon request.

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

## Acknowledgements

We would like to thank the many colleagues who contributed to collection and phenotypic characterization of the clinical samples, as well as genotyping and analysis of the GWAS data. Full details of acknowledgements are provided below.

ALSPAC Study: We are extremely grateful to all the families who took part in this study, the midwives for their help in recruiting them, and the whole ALSPAC team, which includes interviewers, computer and laboratory technicians, clerical workers, research scientists, volunteers, managers, receptionists, and nurses. GWAS data was generated by Sample Logistics and Genotyping Facilities at the Wellcome Trust Sanger Institute and LabCorp (Laboratory Corporation of America) using support from 23andMe. The UK Medical Research Council and the Wellcome Trust (Grant ref: 102215/2/13/2) and the University of Bristol provide core support for ALSPAC. This publication is the work of the authors, and JPK and DME will serve as guarantors for the contents of this paper. JPK is funded by a Wellcome Trust 4-year PhD studentship in molecular, genetic, and life course epidemiology (WT083431MA). This work is supported by a Medical Research Council program grant (MC_UU_12013/4 to D.M.E). D.M.E. is supported by an Australian Research Council Future Fellowship (FT130101709).

The Generation R Study: We gratefully acknowledge the contribution of children and parents, general practitioners, hospitals, midwives and pharmacies in Rotterdam. The generation and management of GWAS genotype data for the Generation R Study was done at the Genetic Laboratory of the Department of Internal Medicine, Erasmus MC, The Netherlands. We thank Pascal Arp, Mila Jhamai, Marijn Verkerk, Lizbeth Herrera and Marjolein Peters for their help in creating, managing and QC of the GWAS database. The musculoskeletal research of the Generation R study is partly supported by the European Commission grant HEALTH-F2-2008-201865-GEFOS. The general design of Generation R Study is made possible by financial support from the Erasmus Medical Center, Rotterdam, the Erasmus University Rotterdam, the Netherlands Organization for Health Research and Development (ZonMw), the Netherlands Organisation for Scientific Research (NWO), the Ministry of Health, Welfare and Sport. Additionally, the Netherlands Organization for Health Research and Development supported authors of this manuscript (ZonMw 907.00303, ZonMw 916.10159, ZonMw VIDI 016.136.361 to V.W.V.J., and ZonMw VIDI 016.136.367 to F.R.)

COPSAC: We express our gratitude to the participants of the COPSAC2000, COPSAC2010 and COPSAC-REGISTRY cohort study for all their support and commitment. We also acknowledge and appreciate the unique efforts of the COPSAC research team. The Bone Mineral Density in Childhood Study: BMDCS is extremely grateful to all the families who participated in this study, and the whole team, which includes interviewers, computer and laboratory technicians, clerical workers, research scientists, volunteers, managers, receptionists, and nurses. This work was funded by the National Institute of Child Health and Human Development (NICHD) contracts NO1-HD-1-3228, −3329, −3330, −3331, −3332 and −3333, R01 HD058886 and the Clinical and Translational Research Center (5-MO1-RR-000240 and UL1 RR-026314).

The CHARGE Consortium: Results from the meta-analysis of GWAS for lean body mass in adults were provided by the Musculoskeletal Working Group of the Cohorts for Heart and Aging Research on Genomic Epidemiology ("CHARGE", http://depts.washington.edu/chargeco/wiki/Main_Page), which is an international consortium of cohort studies focused on a variety of disease phenotypes. Building on GWAS for NHLBI-diseases: the U.S. CHARGE Consortium' was funded by the NIH (the American Recovery and Reinvestment Act of 2009, ARRA, 5RC2HL102419). DPK was supported by a grant from the National institute on Arthritis, Musculoskeletal and Skin Diseases (R01 AR41398).

## Author contributions

Conceived and designed the experiments: C.M.G., J.P.K., J.H.T., D.M.E., and F.R. Performed the experiments: C.M.G., J.P.K., E.K., A.C., C.G.B., T.S.A., D.H.M.H., M.G., S.D., D.P.K., C.A.-B., S.R., J.v.d.P. Analyzed the data: C.M.G., J.P.K., N.L.D., E.K., A.C., C.G.B., E.E., M.G., S.D., D.P.K., B.C.J.v.E., C.A.-B., S.R., J.v.d.P., J.H.T., P.G.B., D.M.E., F.R., Y.-H.H. Contributed reagents/materials/analysis tools: N.L.D., C.G.B., M.G., E.E., B.C.J.v.E., C.A.-B., S.R., K.M.G., T.R., J.v.d.P., P.G.B. Wrote first draft of the paper: C.M.-G., F.R. Contributed and approved final version of the paper: C.M.G., J.P.K., N.L.D., E.K., A.C., B.S.Z., K.B., C.G.B., T.S.A., H.B., E.E., D.H.M.H., L.F.B., J.P.G., M.G., Y.-H.H., S.D., D.G.D., M.T.M., D.P.K., B.C.J.v.d.E., C.A.-B., S.R., K.M.G., T.R., D.K., J.v.d.P., V.W.V.J., A.G.U., J.H.T., S.F.A.G., D.M.E., P.G.B., F.R.
