## [Peer Review File · Nature Communications]

Reviewers' comments:

Reviewer #1 (Remarks to the Author):

The authors have undertaken a bivariate GWAS of BMD and lean mass in children and identified known bone loci and one novel locus. This is an interesting paper and may help to better understand the genetics of lean mass, which has provide difficult to date.

Of course the major limitation of this paper is that it picked up only hits that were GWS for BMD, or nearly, except or TOM1L2 locus. This locus does not seem to be driven only by BMD, however, there is no independent replication of these findings. Could this not be done using data from adult BMD and LM GWAS?

The overall genetic correlations of these two traits is positive, not negative. However, the lead SNP rs7501812 has an opposite direction of effect in TB BMD and LM. This is not well emphasized.

It would be a lot easier reading if the authors described the known antagonistic effects of SREBF1 on osteoblast and myoblast differentiation in the results.

The consistent pattern of directionality across traits in adult populations for the rs7501812 is not terribly convincing for FN BMD and Appendicular Lean mass, given the weak p-values. I would change this sentence in the manuscript.

I am very confused by this sentence: "None of these genes revealed expression profiles in bone tissue that correlated with donor phenotypes". I thought that SREBF1 was negatively correlated with FN BMD?

A diagram outlining the authors thoughts on the effect of the allele at rs7501812 on expression, BMD and LM would be very helpful for the readership.

Other Points

1. Rather than stating >10,000 children in the abstract, why not just list the number?
Please provide data showing that DXA derived lean mass is a good proxy for skeletal mass, rather than just citing articles.
2. Several GWAS of BMD are excluded in references, including Icelandic GWAS and the UK10K-based GWAS.
3. Please add loci/gene column names to Table 3.
4. Supplementary Tables in Excel are way more helpful than PDFs!
5. There is no reference for the cis-eQTL effect in whole blood.

Reviewer #2 (Remarks to the Author):

In this manuscript, Medina Gomez et al. reported a bivariate genome-wide association study (GWAS) for total body bone mineral density (TB-BMD) and lean mass (TB-LM) in children. The authors

estimated heritability and genetic correlation for TB-BMD and TB-LM, and performed univariate and bivariate GWAS for these two traits. The bivariate GWAS detected significant association for TB-BMD and TB-LM at eight genomic loci, including a few loci not showing genome-wide significance in univariate analysis. The authors then focused subsequent bioinformatics and functional analyses on the locus at 17p11.2, which was not associated either trait in univariate analyses. By interrogating a diverse collection of functional data, such as eQTL data in whole blood and skeletal muscle, chromatin interaction pattern in oncogenic cell lines, and gene expression during in vitro osteoblastogenesis, the authors suggested that genes SREBF1/TOM1L2 are the main candidate genes underlying the observed bivariate association signal. This study was largely appropriately conducted and provided valuable insights into the shared genetic basis of BMD and LM. However, a few issues need to be addressed.

Major issues:

1. The authors should provide further justification for using TB-BMD and TB-LM as the study phenotypes. First, it is generally accepted that the genetic determinants for BMD at different skeletal sites may be different, and thus genetic association studies were normally conducted for site-specific BMDs. In addition, appendicular LM may be a better proxy for skeletal muscle mass than TB-LM, because TB-LM measured by DXA also includes measurements from organ tissues.
2. Although the authors applied extensive bioinformatics and functional analyses, however, the functional significance of the SREBF1 and TOM1L2 genes are still murky. For example, the supporting evidences were largely based on the eQTL data and chromatin interaction data in non-musculoskeletal cells. Also, the expression of these two genes in female thigh muscle did not show significant correlation for both BMD and muscle/lean mass phenotypes, and the knockout mice for these two genes have no skeletal phenotypes. The authors should have further discussion on this issue.
3. A result comparison and discussion about the previously reported bivariate GWAS for appendicular LM and bone size (Guo YF, Hum Genet, 2013), and bivariate whole-genome linkage analysis between TB-LM and BMD (Wang XL, JBMR, 2007) should be added.

Minor issues:

4. On page 7, line 198, it seems that the "Supplementary Table 3" should be "Supplementary Table 5".
5. The figure legend for Supplementary Figure 5 should indicate the data is for SREBF1 gene.

REVIEWERS' COMMENTS:

Reviewer #1 (Remarks to the Author):

The authors have addressed my concerns.

Reviewer #2 (Remarks to the Author):

The authors have carefully addressed all previous reviewers' critiques and I have no further concerns.

Response to reviewers:

We would like to thank the editor for the interest and both reviewers for the appreciation of our work. We are also thankful to the reviewer for their remarks and comments, which we have incorporated in this new version of our manuscript to the best of our abilities. The specific responses and changes to our manuscript are described below. In order to incorporate all of the suggested changes while keeping the length of the manuscript unchanged, we have opted to move the description of all loci other than *TOM1L2/SREBF1* to the Supplementary Note, section “Description of TBLM/TBLH-BMD associated loci”.

Reviewer #1 (Remarks to the Author):

The authors have undertaken a bivariate GWAS of BMD and lean mass in children and identified known bone loci and one novel locus. This is an interesting paper and may help to better understand the genetics of lean mass, which has provide difficult to date.

1. Of course the major limitation of this paper is that it picked up only hits that were GWS for BMD, or nearly, except or TOM1L2 locus. This locus does not seem to be driven only by BMD, however, there is no independent replication of these findings. Could this not be done using data from adult BMD and LM GWAS?

Based on the current findings of our pediatric meta-analysis we are launching an initiative to carry on the same bivariate analysis of lean mass and BMD in adults, within the setting of the GEFOS consortium, targeting a sample size of more than 100,000 participants. Such large sample size setting is probably required for the genetic analysis in adults, as indicated by our previous analysis on BMD¹, and results in other complex traits also largely influenced by environmental factors like blood pressure² and BMI³. Given the large scale of this new project results will not be available in the near future. Nevertheless, to address this point we used the summary statistics from the discovery sets of BMD⁴ (N=33,000), measured both at the lumbar spine and femoral neck, and whole body lean mass⁵ (N=38,300) meta-analyses drawn in adults and elderly individuals. We performed a bivariate analyses using the empirical-weighted linear-combined test statistics (eLC) approach^{6,7} (described in <https://sites.google.com/site/multivariateyihsianghsu/>). We outline the approach in the

Methods section on page 20 (lines 438-452). We have added these bivariate results to the univariate results previously presented in **Supplementary Table 4** only for the 17p11.2 region. In the Results section we have added the following paragraph on page 7 (lines 148-154): “Additionally, we applied a bivariate analysis to the summary statistics of previously reported univariate GWAS meta-analyses of BMD and lean mass traits in adults^{21,28} (Supplementary Table 4). We only found evidence of genetic variants exerting pleiotropic effects on both traits in the 11q13.2 locus. However, consistent with our findings in children, variants in the 17p11.2 locus (led by rs7501812) showed opposite association with lumbar spine BMD and TB-LM at the margin of reaching statistical significance ($P < 0.07$).” As suggested by reviewer 2 (see point 3), we have added to the discussion on page 13 (lines 284-289) a paragraph making reference to other published bivariate efforts: “Our meta-analysis did not replicate potential pleiotropic signals previously reported in adults^{33,34}. Nevertheless, the bivariate analysis of summary statistics of published BMD and lean mass efforts in adults we did, identified variants with strong evidence for pleiotropic effects in the 11q13.2 locus. The lack of replication in the other regions may well be a consequence of differential genetic effects during the life course, e.g., the accumulation of environmental factors attenuating effects in the elderly.”

2. The overall genetic correlations of these two traits is positive, not negative. However, the lead SNP rs7501812 has an opposite direction of effect in TB BMD and LM. This is not well emphasized.

We have emphasized this fact in the abstract, as well as in the results section in page 6, lines 138-139: “All GWS SNPs in this region yielded nominally significant opposite effect for the coded allele in TBLH-BMD as compared to TB-LM, despite the positive correlation between these traits.”; and also in the discussion in page 13 lines 296-298 highlighting that their discovery may be likely facilitated by the increased power of the bivariate analysis under the scenario of opposite effects: “Despite the positive correlation between TB-LM and TBLH-BMD, variants in this region exerted opposite effects on these traits, probably facilitating their discovery in a bivariate analysis powerful setting”.

3. It would be a lot easier reading if the authors described the known antagonistic effects of SREBF1 on osteoblast and myoblast differentiation in the results.

As suggested by the reviewer we have moved the paragraph dedicated to the opposing effects of *SREBF1* on osteoblast and myoblast differentiation to the results section. Therefore, now on page 10 (lines 221-230) the text reads “*SREBF1* is an adipocyte differentiation factor (ADD-1) that produces SREBP-1, a transcription factor ubiquitously expressed (more strongly in lipogenic tissues) and directly regulating the transcription of over 200 genes involved in the *de novo* synthesis of fatty acids, triglycerides, and cholesterol⁴⁴. SREBP-1, in its active form, is important for the mineralization of osteoblastic cultures *in vitro*, as its overexpression increases the number of mineralized foci⁴⁵. Contrary to its positive regulatory role in osteoblast differentiation and mineralization, in skeletal muscle SREBP-1 protein indirectly downregulates *MYOD1*, *MYOG* and *MEF2C*, acting as a key regulator of myogenesis. Similarly, overexpression of SREBP-1 inhibits myoblast-to-myotube differentiation⁴⁶, reduces cell size and leads to loss of muscle-specific proteins in differentiated myotubes⁴⁴”.

4. The consistent pattern of directionality across traits in adult populations for the rs7501812 is not terribly convincing for FN BMD and Appendicular Lean mass, given the weak p-values. I would change this sentence in the manuscript.

Following this suggestion and the results obtained from the bivariate analysis in adults (see point 1) we have rephrased this statement in the results section page 7 (lines 148-154), “Additionally, we applied a bivariate analysis to the summary statistics of previously reported univariate GWAS meta-analyses of BMD and lean mass traits in adults^{21,28} (**Supplementary Table 4**). We only found evidence of genetic variants exerting pleiotropic effects on both traits in the 11q13.2 locus. However, consistent with our findings in children, variants in the 17p11.2 locus (led by rs7501812) showed opposite association with lumbar spine BMD and TB-LM at the margin of reaching statistical significance ($P < 0.07$).”

5. I am very confused by this sentence: “None of these genes revealed expression profiles in bone tissue that correlated with donor phenotypes”. I thought that *SREBF1* was negatively correlated with FN BMD?

We apologize for the misunderstanding. We have expression profiling and DXA measurements available for two different sets of women. The first set consisted of expression profiling in human thigh muscle biopsies from 18 healthy postmenopausal women; in this set high expression of *SREBF1* (in muscle) was correlated with low FN-BMD ($P=10^{-4}$). The second set consisted of expression profiling in bone obtained through trans-iliac bone biopsies from 84 postmenopausal women distributed across the normal, osteopenia and osteoporosis spectrum. In the latter set, we identified no significant correlation between gene expression profiles of any of the transcripts in *SREBF1* or the contiguous genes (*RAI1*, *TOM1L2* and *ATPF2*), and any of the DXA parameters.

We have rewritten this paragraph of the Results section on page 10 (lines 233-246) to better describe the distinction between the two expression profiling sets. “*SREBF1* muscle expression, assessed from postmenopausal women thigh muscle biopsies (N=18) showed significant negative correlation with femoral neck BMD of the donor ($P<0.001$) and was borderline significantly associated with whole body BMD ($P=0.05$) (**Supplementary Table 6**). Expression of *RAI1* was inversely correlated with thickness of the *vastus lateralis* muscle of the donor ($P=0.01$), while *TOM1L2* expression levels were positively correlated with this trait ($P=0.02$). *ATPAF2* and *C17orf39* [*GID4*] did not correlate with any of the tested measurements of the donors. Evaluation of expression profiles from trans-iliac bone biopsies in a separate group of postmenopausal women (N=80), revealed no correlation ($P<0.05$) with donor phenotypes for either of the genes despite being expressed in bone [data not shown].” Also, we have moved the table containing these results to the Supplementary Material (see also point 2 of reviewer 2)

6. A diagram outlining the authors thoughts on the effect of the allele at rs7501812 on expression, BMD and LM would be very helpful for the readership.

Following the reviewer’s suggestion, we have added a diagram of the hypothesis we derived based on our observed results and previous reports on the role of *SREBF1* in osteoblast and myoblast differentiation, also shown below. We placed this figure in the discussion as it integrates the different levels of evidence acknowledging those components where the presented evidence is insufficient. We leave at discretion of the editor to include (or not) this figure in the main article.

Figure 5. Schematic representation of the plausible role of rs7501812 in the pleiotropic modulation of bone density and lean mass. The G-allele from rs7501812 upregulates the expression of *SREBF1* both in skeletal muscle and bone. This overexpression would be expected to result in higher levels of the active form of SREBP-1. SREBP-1 exerts opposite effects on bone and muscle biogenesis. While it promotes osteoblast mineralization⁴⁵ it inhibits myoblast differentiation^{44,46}. Ultimately, this modulation would result in higher BMD and lower lean mass, as we observed in our bivariate GWAS analysis.

Other Points

7. Rather than stating >10,000 children in the abstract, why not just list the number?

The exact number of participants is now presented in the abstract.

8. Please provide data showing that DXA derived lean mass is a good proxy for skeletal mass, rather than just citing articles.

The reviewer raises an important point. Unfortunately, we do not have a “gold standard” technique such as MRI or CT, which can measure muscle mass with which to compare the lean mass determined by DXA. However, following the reviewer’s concern, we not only cite the

articles, but also present on page 3 (lines 73-78) a detailed description of the high correlation between DXA-derived lean mass and muscle mass measured with these techniques. The new text reads “Lean mass, which is a good proxy for skeletal muscle mass^{10,11}, can also be derived from the same whole-body DXA scans. As shown by Chen et al. in postmenopausal women¹⁰, DXA lean mass has a high correlation ($\rho=0.94$) with skeletal muscle mass of the whole body measured by magnetic resonance imaging (MRI). Furthermore, Bridge et al. found in peripubertal children¹¹ a 0.98 correlation between DXA derived lean mass in the mid third femur and skeletal muscle mass of the same region measured with MRI”.

9. Several GWAS of BMD are excluded in references, including Icelandic GWAS and the UK10K-based GWAS.

We thank the reviewer for pointing out this omission. We have added four references to the paper: the three recent deCODE studies implicating variants in *PTCH1*⁸, *COL1A2*⁹, and *LGR4*¹⁰, together with a multistage meta-analysis implicating *SMOC1* and *CLDN14*¹¹, loci that have not been identified in any of the already cited studies.

10. Please add loci/gene column names to Table 3.

The loci information has been added to Table 3

11. Supplementary Tables in Excel are way more helpful than PDFs!

We agree with the reviewer. Excel files will then be provided for long tables.

12. There is no reference for the cis-eQTL effect in whole blood.

We thank the reviewer for noticing this omission. The reference was provided later in the text in the SMR analysis methods and results sections, but indeed the reference was lacking when reporting the primary eQTL results. The reference has now been added.

Reviewer #2 (Remarks to the Author):In this manuscript, Medina Gomez et al. reported a bivariate genome-wide association study (GWAS) for total body bone mineral density (TB-BMD) and lean mass (TB-LM) in children. The authors estimated heritability and genetic correlation for TB-BMD and TB-LM, and performed univariate and bivariate GWAS for these two traits. The bivariate GWAS detected significant association for TB-BMD and TB-LM at eight genomic loci, including a few loci not showing genome-wide significance in univariate analysis. The authors then focused subsequent bioinformatics and functional analyses on the locus at 17p11.2, which was not associated either trait in univariate analyses. By interrogating a diverse collection of functional data, such as eQTL data in whole blood and skeletal muscle, chromatin interaction pattern in oncogenic cell lines, and gene expression during in vitro osteoblastogenesis, the authors suggested that genes SREBF1/TOM1L2 are the main candidate genes underlying the observed bivariate association signal. This study was largely appropriately conducted and provided valuable insights into the shared genetic basis of BMD and LM. However, a few issues need to be addressed.

Major issues:

1. The authors should provide further justification for using TB-BMD and TB-LM as the study phenotypes. First, it is generally accepted that the genetic determinants for BMD at different skeletal sites may be different, and thus genetic association studies were normally conducted for site-specific BMDs. In addition, appendicular LM may be a better proxy for skeletal muscle mass than TB-LM, because TB-LM measured by DXA also includes measurements from organ tissues.

We thank the reviewer for bringing this important point up. It is true that in adults it is common clinical practice to measure BMD at the sites where fracture is more common (i.e., hip, lumbar spine and forearm). On the other hand, for the assessment of bone health in children, from infancy to adolescence, total body less head (TBLH-BMD) is one of the preferred sites for measurement according to the International Society for Clinical Densitometry¹². The reason is that changes in bone area due to growth create artifacts influencing the reproducibility, comparability and interpretation of DXA measurements, and this bias is accentuated when measuring BMD in smaller areas. As all the cohorts in our study are pediatric, we used TBLH-

BMD to measure bone density and coupled it to lean mass measured also at the whole body in the same scans.

We agree with the reviewer that appendicular lean mass could be a better proxy for skeletal muscle mass. Nonetheless, appendicular lean mass is largely environmentally determined what could affect the power to detect genetic variants influencing the process of muscle gain. We compared total body and appendicular lean mass within the largest cohorts participating in the analysis - Avon Longitudinal Study of Parents and their Children ((ALSPAC), N=5,251) and The Generation R Study (N=4,071) - . The phenotypic correlation between appendicular and total body lean mass was over 0.92 in both cohorts.

2. Although the authors applied extensive bioinformatics and functional analyses, however, the functional significance of the SREBF1 and TOM1L2 genes are still murky. For example, the supporting evidences were largely based on the eQTL data and chromatin interaction data in non-musculoskeletal cells. Also, the expression of these two genes in female thigh muscle did not show significant correlation for both BMD and muscle/lean mass phenotypes, and the knockout mice for these two genes have no skeletal phenotypes. The authors should have further discussion on this issue.

We thank the reviewer for expressing this concern likely consequence of our intention to put together a comprehensive functional evaluation. As suggested by Reviewer 1 (see figure in point 6 of reviewer 1), we have attempted to summarize the different levels of evidence and proposed a working hypothesis of the underlying mechanisms, but have also acknowledged the components where the functional evidence is insufficient. On top of the evidence derived from the eQTL and chromatin interaction data in non-musculoskeletal cells, previous functional experiments carried out independently on muscle and bone have proven the opposing role of SREBP-1 (the product from *SREBF1*) on both osteoblast and myoblast differentiation. Further, these studies are directionally consistent with our GWAS findings and working hypothesis. Indeed, in contrast to other cell and tissues, functional evaluations in musculoskeletal cells are scarce or not readily available. Despite the fact that some of the presented functional evaluations arise from relatively small studies in a non-pediatric setting (i.e., muscle/bone

expression profiling) and not necessarily in line with our working hypothesis, we have opted to include them on the paper to avoid selective reporting.

We have also completed and clarified the information summarized in **Supplementary Table 3**. There are no *Srebf1* or *Tom1l2* knockout models screened by the International Mouse Phenotyping Consortium (www.mousephenotype.org/). Nevertheless, there are indeed few transgenic models for *Srebf1* described in the literature including two knockouts^{13 14}, one GeneTrap¹⁵ and an over-expression model¹⁶. None of the published reports of these transgenics describe skeletal phenotyping; rather the phenotypic reports from these strains present assessments only on serum lipids and hepatic measurements. As such, we cannot determine if a skeletal phenotype was never screened or if it was absent. In contrast, we became aware of one *Tom1l2* GeneTrap model reporting kyphosis and malocclusion of teeth¹⁷. Therefore, in the **Supplementary Table 3**, we now state for *Srebf1* “no information about skeletal phenotype has been reported”, whereas for *Tom1l2* “Kyphosis” has been reported. We have further clarified this point in the discussion page 15 lines (329-332) “Our *ex-vivo* models in murine osteoblasts established that *Rai1*, *Atpaf2*, *Srebf1* and *Tom1l2* were expressed in bone. With the exception of *Srebf1*, knockout mice for these genes all present with a skeletal phenotype. While *Srebf1* KO models have been reported in the literature, we have no certainty that bone phenotyping was performed on these mice.”

Regarding the female thigh muscle biopsies, the *SREBF1* expression is correlated with lower FN-BMD ($P=7 \times 10^{-4}$) and lower TB-BMD ($P=0.05$) (**Supplementary Table 6**). Yet, the muscle biopsies are from a small number of post-menopausal women (N=18), in which we also failed to identify significant correlations with lean and fat mass, although *SREBF1* is a key regulator of adipogenesis and lipid/cholesterol synthesis. Considering that these correlations should be interpreted with caution, we have opted to tone down the statements derived from this analysis, exclude this information from the abstract and move the results to the supplement (**Supplementary Table 6**). We now state in the discussion (page 16, 350-356): “Whereas, the expression of *SREBF1* in muscle biopsies correlated with lower BMD parameters in a direction

opposite to what is predicted by the SREBP-1 function in bone and muscle metabolism, the number of expression profiles included in analysis is small; also, this unexpected correlation could arise from tissue-specific effects.”

3. A result comparison and discussion about the previously reported bivariate GWAS for appendicular LM and bone size (Guo YF, Hum Genet, 2013), and bivariate whole-genome linkage analysis between TB-LM and BMD (Wang XL, JBMR, 2007) should be added.

We have followed the suggestion and looked into the replication of the findings from these two studies in our GWAS. As shown in the regional plots at the end of this document, none of the reported suggestive signals show evidence for replication in our study. To note, Guo et al.¹⁸ used a measure of bone size at the hip and appendicular lean mass in their study, and their only GWS variant was of low frequency (MAF 0.02) therefore, it was not part of our study including variants with MAF \geq 0.05. The study of Wang et al.¹⁹, performed a bivariate analysis of BMD measured at specific skeletal sites and total body lean mass, and only identified sex-specific associations. There are several differences in the previously published studies and in our own study that should be highlighted. Our study examined total body BMD and lean mass in pediatric cohorts, and analyzed boys and girls together. As such, only genes exerting effects pre-pubertally could be captured by our bivariate approach. We have included a paragraph in the results section on page 6 (lines 140-146), where we also refer to an extra analysis performed using summary data from univariate meta-analysis of femoral neck and lumbar spine BMD and lean mass (see point 1 from reviewer 1). “Although using different phenotypes than the ones used in our analysis, GWAS bivariate strategies in adult populations have approached the bone/lean mass relationship. A GWAS bivariate analysis of bone size and appendicular lean mass in Chinese and European individuals³³ reported a potential association signal mapping to the *GLYAT* gene (11q12.1) arising from a low frequency (MAF<0.05) variant not present in our meta-analysis. In addition, a linkage study reported a significant signal (LOD score =4.86) mapping to the 15q13 locus and multiple suggestive signals (LOD score < 3) in 7p22, 7q21, 7q32 and 13q11³⁴. In our bivariate GWAS meta-analysis none of these regions (including *GLYAT*) contained significantly associated SNPs ($P < 9.0 \times 10^{-6}$ after multiple correction,

5,537 SNPs tested)". And also in the discussion, page 13 lines 284-289 "Our meta-analysis did not replicate potential pleiotropic signals previously reported in adults^{33,34}. Nevertheless, the bivariate analysis of summary statistics of published BMD and lean mass efforts in adults we did, identified variants with strong evidence for pleiotropic effects in the 11q13.2 locus. The lack of replication in the other regions may well be consequence of differential genetic effects during the life course, e.g. the accumulation of environmental factors attenuating effects in the elderly."

Minor issues:

4. On page 7, line 198, it seems that the "Supplementary Table 3" should be "Supplementary Table 5".

The reviewer is right. The change has been made.

5. The figure legend for Supplementary Figure 5 should indicate the data is for SREBF1 gene.

We have added the gene to the caption.

References

1. Kemp, J.P., Medina-Gomez, C., Tobias, J.H., Rivadeneira, F. & Evans, D.M. The case for genome-wide association studies of bone acquisition in paediatric and adolescent populations. *Bonekey Rep* **5**, 796 (2016).
2. Parmar, P.G. *et al.* International GWAS Consortium Identifies Novel Loci Associated with Blood Pressure in Children and Adolescents. *Circ Cardiovasc Genet* (2016).
3. Winkler, T.W. *et al.* The Influence of Age and Sex on Genetic Associations with Adult Body Size and Shape: A Large-Scale Genome-Wide Interaction Study. *PLoS Genet* **11**, e1005378 (2015).
4. Estrada, K. *et al.* Genome-wide meta-analysis identifies 56 bone mineral density loci and reveals 14 loci associated with risk of fracture. *Nat Genet* **44**, 491-501 (2012).
5. Zillikens, M.C. *et al.* Large Meta-Analysis of Genome Wide Association Studies Identifies Five Loci for Lean Body Mass. (2016).
6. Ligthart, S. *et al.* Bivariate genome-wide association study identifies novel pleiotropic loci for lipids and inflammation. *BMC Genomics* **17**, 443 (2016).
7. Perry, J.R. *et al.* DNA mismatch repair gene MSH6 implicated in determining age at natural menopause. *Hum Mol Genet* **23**, 2490-7 (2014).
8. Styrkarsdottir, U. *et al.* Sequence variants in the PTCH1 gene associate with spine bone mineral density and osteoporotic fractures. *Nat Commun* **7**, 10129 (2016).
9. Styrkarsdottir, U. *et al.* Two Rare Mutations in the COL1A2 Gene Associate With Low Bone Mineral Density and Fractures in Iceland. *J Bone Miner Res* **31**, 173-9 (2016).
10. Styrkarsdottir, U. *et al.* Nonsense mutation in the LGR4 gene is associated with several human diseases and other traits. *Nature* **497**, 517-20 (2013).
11. Zhang, L. *et al.* Multistage genome-wide association meta-analyses identified two new loci for bone mineral density. *Hum Mol Genet* **23**, 1923-33 (2014).
12. Lewiecki, E.M. *et al.* Special report on the 2007 adult and pediatric Position Development Conferences of the International Society for Clinical Densitometry. *Osteoporos Int* **19**, 1369-78 (2008).
13. Shimano, H. *et al.* Elevated levels of SREBP-2 and cholesterol synthesis in livers of mice homozygous for a targeted disruption of the SREBP-1 gene. *J Clin Invest* **100**, 2115-24 (1997).
14. Liang, G. *et al.* Diminished hepatic response to fasting/refeeding and liver X receptor agonists in mice with selective deficiency of sterol regulatory element-binding protein-1c. *J Biol Chem* **277**, 9520-8 (2002).
15. Im, S.S. *et al.* Sterol regulatory element binding protein 1a regulates hepatic fatty acid partitioning by activating acetyl coenzyme A carboxylase 2. *Mol Cell Biol* **29**, 4864-72 (2009).
16. Shimomura, I. *et al.* Insulin resistance and diabetes mellitus in transgenic mice expressing nuclear SREBP-1c in adipose tissue: model for congenital generalized lipodystrophy. *Genes Dev* **12**, 3182-94 (1998).
17. Girirajan, S. *et al.* Tom1l2 hypomorphic mice exhibit increased incidence of infections and tumors and abnormal immunologic response. *Mamm Genome* **19**, 246-62 (2008).
18. Guo, Y.F. *et al.* Suggestion of GLYAT gene underlying variation of bone size and body lean mass as revealed by a bivariate genome-wide association study. *Hum Genet* **132**, 189-99 (2013).
19. Wang, X.L. *et al.* Bivariate whole genome linkage analyses for total body lean mass and BMD. *J Bone Miner Res* **23**, 447-52 (2008).

Appendix

Regional plots of previously reported suggestive signals in our bivariate analysis

15q13

Wang et al identified the 15q13loci as GWS associated with TBLM and spine BMD in women

13q11

Wang et al identified suggestive evidence for association for TBLM and BMD at both spine and hip in women and 13p11

7q32

Wang et al identified suggestive evidence for association for TBLM and BMD at both spine and hip in women and 7q32

7p22

Wang et al identified suggestive evidence for association TBLM and spine BMD for the entire sample and 7p22

7q21

Wang et al identified suggestive evidence for association for TBLM and BMD at both spine and hip in women and 13p11

11q12.1 (GLYAT)

Guo et al. identified a low frequency variant in 11q12.1 (GLYAT) as associated with appendicular lean mass and appendicular bone size.